

# Overland movement in African clawed frogs (*Xenopus laevis*): a systematic review

John Measey

Centre for Invasion Biology, Department of Botany & Zoology, Stellenbosch University, South Africa

## ABSTRACT

African clawed frogs (*Xenopus laevis*) are often referred to as 'purely aquatic' but there are many publications which suggest extensive overland movements. Previous reviews which considered the topic have not answered the following questions: (1) is there evidence for overland dispersal in native and invasive ranges; (2) what is the range of distances moved overland; (3) when does overland movement occur; and (4) is there evidence of breeding migratory behaviour? A systematic review was chosen to synthesise and critically analyse all literature on the overland movement in *Xenopus laevis*. Database searches resulted in 57 documents which revealed a paucity of empirical studies, with 28 containing no data, and 19 having anecdotal content. Overwhelming evidence shows that both native and invasive populations of *X. laevis* move overland, with well documented examples for several other members of the genus (*X. borealis, X. gilli, X. muelleri, X. fraseri* and *X. tropicalis*). Reports of distances moved overland were from 40 m to 2 km, with no apparent difference between native and invasive ranges. Overland movements are not confined to wet seasons or conditions, but the literature suggests that moving overland does not occur in the middle of the day. Migrations to temporary water-bodies for breeding have been suggested, but without any corroborating data.

## INTRODUCTION

Dispersal is a key trait in the life-history of any organism influencing the distribution, community structure and abundance of populations (*Clobert et al., 2009*). In anthropogenically disturbed environments, dispersal may be interrupted or facilitated by novel landscape features that may hinder the conservation of threatened species or facilitate the spread of invasive species (*Carr & Fahrig, 2001*; *Brown et al., 2006*). For invasive species, dispersal is one of the main variables which determines the success of establishment, as well as the rate of spread (*Wilson et al., 2009*). In fact, dispersal affects some aspects of all ecological, evolutionary and conservation problems. Amphibians are model organisms for studies in dispersal as they are generally thought to have low dispersal abilities which brings about strong phylogeographical structuring (e.g., *Avise, 2000*).

Despite their reputation for strong site fidelity, amphibians have been shown to have considerable dispersal abilities. *Smith & Green (2005)* reviewed evidence for maximum

Corresponding author
John Measey, john@measey.com

dispersal in amphibians and concluded that although most individual anurans move short distances (<1 km), small numbers of individuals could be expected to move much further (>10 km). Moreover, these dispersal events may well be informed by a multisensory orientation system that enables individuals to locate water-bodies in which to complete their complex life-histories (*Sinsch, 2006*). For most amphibians, this involves laying eggs into water where larvae grow and metamorphose to emerge onto land. But for frogs in the genus *Xenopus*, adults inhabit the same water as their eggs and larvae, prompting many workers to refer to them as 'completely' or 'purely' aquatic (e.g., *Elepfandt et al., 2000*).

The African clawed frog, *Xenopus laevis*, is one of four model vertebrate species (*Travis, 2006*), and as such has been distributed to laboratories globally (*Van Sittert & Measey, 2016*), as well becoming very popular in the pet trade (*Measey , in press*). This has resulted in invasive populations on four continents (*Measey et al., 2012*) and the suggestion that climate-change may increase invasion success in Europe (*Ihlow et al., 2016*). Surprisingly, the ecology of *X. laevis* is better studied in invasive populations than in their native range, and this lack of ecological data from the native range is problematic as it stymies interpretation of invasive studies. Data on overland movement is particularly important for this principally aquatic amphibian, as it provides insights into dispersal and thus invasion potential.

There is no doubt that *Xenopus laevis*, like other species in the genus *Xenopus* and the family Pipidae, are secondarily aquatic (*Gans & Parsons, 1966*; *Trueb, 1996*), spending the majority of their active time within water-bodies. They have a number of morphological and anatomical adaptations to an aquatic lifestyle including an extensive lateral line system in adults (*Elepfandt, 1996*), aquatic olfactory receptors (*Freitag et al., 1995*), type I ilio-sacral articulation for more efficient swimming locomotion (*Emerson, 1979*), aquatic auditory apparatus (*Elepfandt et al., 2000*) and suction feeding (*Carreño & Nishikawa, 2010*). However, referring to the species as 'purely aquatic' appears to exclude the possibility of individuals leaving a water-body and travelling overland. It is noteworthy therefore that *X. laevis* retains many traits which have terrestrial functionality, including the auditory apparatus (*Katbamna et al., 2006*; *Mason, Wang & Narins, 2009*), the olfactory apparatus (*Freitag et al., 1995*), and terrestrial jumping and feeding (*Measey, 1998b*). This indicates that terrestrial activities in *X. laevis* are sufficiently important for these animals to have retained terrestrial functions in addition to aquatic specialisations. Although an alternative explanation is that this could be phylogenetic inertia as their ancestors were terrestrial.

Existing literature on overland movement in *Xenopus laevis* dates back to anecdotal observations at the beginning of the twentieth century (*Hewitt & Power, 1913*). However, such records do not appear to agree on whether movements are migrations (to and fro movements of animals between sites; *Hey, 1949*) or animals moving out of drying ponds *en masse* (*Loveridge, 1953*). On the other hand, there appears to be a paucity of direct evidence, with some authors using inferential evidence of overland movement between isolated ponds. Therefore, I conducted a systematic review (see *Moher et al., 2009*) using an *a priori* search strategy and synthesis of all literature on overland movements in African clawed frogs (*Xenopus laevis*) in order to answer the following questions: (1) What is the evidence in the literature for overland dispersal in native and invasive ranges;

(2) What distances are moved overland; (3) When it occurs, is there evidence that overland movement is seasonal or associated with rain or drying habitats; (4) Is there evidence of overland movement being migratory with respect to breeding?

## MATERIALS & METHODS

### Study species

The African clawed frog, *Xenopus laevis*, has undergone significant taxonomic revision following a comprehensive molecular study by *Furman et al. (2015)*. The results of this revision mean that what was previously known as *X. l. laevis* by a number of authors (e.g., *Kobel, Loumont & Tinsley, 1996*; *Poynton, 1964*) is now known as *X. laevis* with all other subspecies being recognised as full species, as well as some newly described species (e.g., *Evans et al., 2015*). The full range of *X. laevis* is now known to cover much of southern Africa: South Africa, Lesotho, Swaziland, Namibia, parts of Botswana, Zimbabwe, parts of Mozambique and extending north into Malawi. While *X. laevis* was the focus of this review, publications that mentioned other species were included and form an integral part of the citation matrix.

### Literature review

A literature search was conducted in Web of Science using the scientific name and derivatives of all common names for *Xenopus laevis* (''clawed frog,'' ''clawed toad'' and ''platanna'') AND (the Boolean search term to stipulate that the record should contain this AND the next term) four terms related to movement overland (''overland,'' ''terrestrial,'' ''migration'' and ''dispersal''). This produced a total of 16 search terms of each pair with 988 results (Table S1). These were then searched for papers in which information on overland movement in *X. laevis* was mentioned, resulting in a total of nine unique papers. An additional search for the words ''*Xenopus* overland'' (appearing anywhere within a document, and for all years) was made in Google Scholar. Google Scholar has an advantage over other literature databases in that the search term may occur anywhere in the text, instead of just in the title, abstract or keywords. This produced 323 results. Each result was inspected to determine whether or not it contained information on the subject matter. Articles that had no relevance (e.g., author was called Overland or subject was not a pipid) were excluded. The remaining articles ($n = 41$) included all nine from the Web of Science search. Google Scholar results were scrutinised for mention of *Xenopus* moving overland. Publications where the subjects *Xenopus* and movement overland were disassociated were removed ($n = 5$). If no evidence was provided but a citation was given, the paper was retained and any citation accessed. Articles that had been cited as giving evidence that *Xenopus* move overland were retained whether or not they actually contained any pertinent information. Citations provided 16 more documents. Lastly, expert knowledge was used to access a further five documents that did not appear in Google Scholar or in citations. This gave a total of 57 documents (Appendix S1). This collection was biased for literature that had electronic full texts that could be crawled by Google Scholar. The additional documents added through citations and by expert knowledge only partially alleviated this bias. Each document was read critically for the information that it contained on *Xenopus*
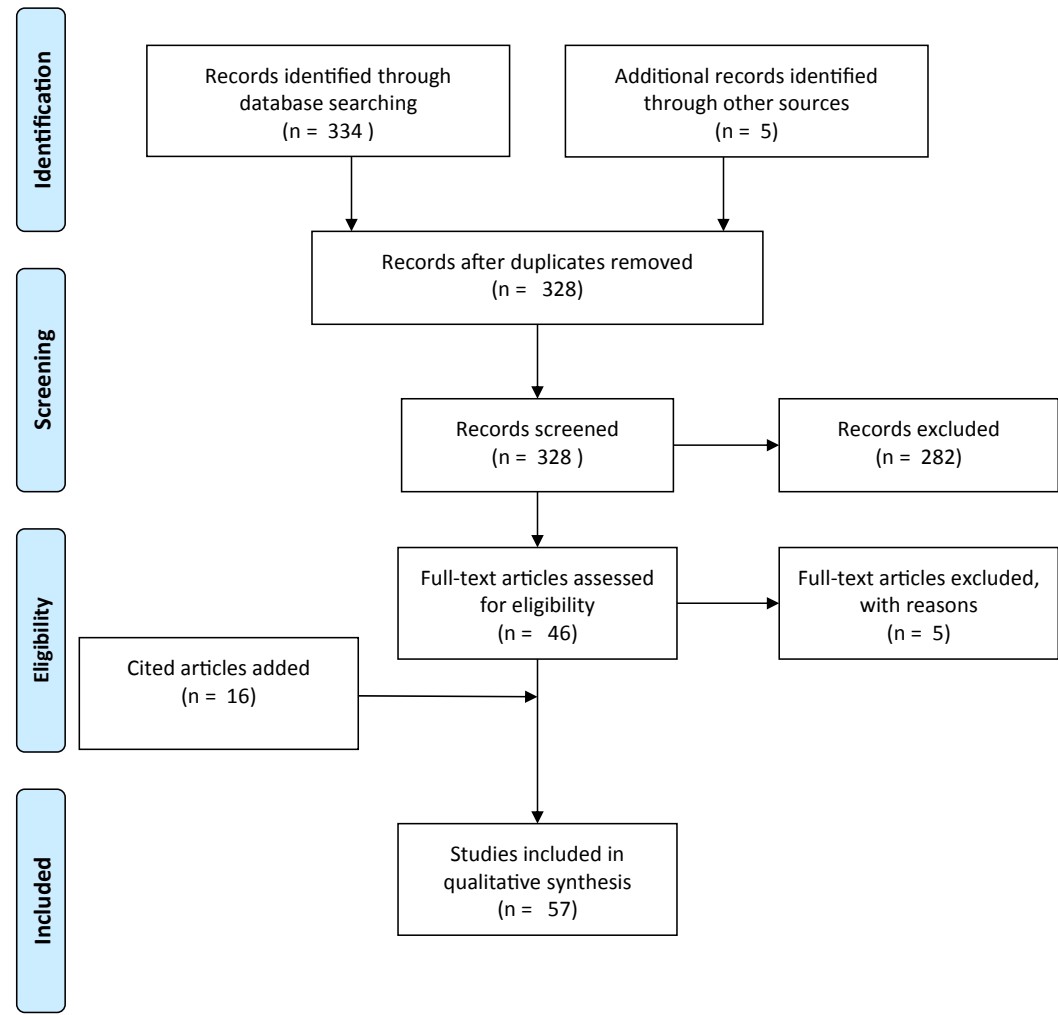

**Figure 1** **Prisma flow-diagram (see** *Moher et al., 2009*) **for literature included in this study.** Flow-diagram for literature on *Xenopus* overland movement included in this study.

moving overland, the species concerned, and with special reference to answering the four study questions. Figure 1 shows a flow diagram for the systematic review following Prisma guidelines (*Moher et al., 2009*).

## Network visualisation

A network visualisation was constructed using Gephi (v 0.8.2) with the aim of showing how citations follow different data types. Literature in the final dataset was classified into five data types: anecdotal (observational reports of frogs moving in terrestrial habitats: $n = 19$), distributional (occurrence of frogs in water bodies separated by terrestrial habitats from potential source populations: $n = 4$), mark-recapture (tagged animals located in water bodies separated by terrestrial habitats from those where they had been marked, including a telemetric study: $n = 4$), reviews ($n = 4$) and publications without any relevant data, but that typically cited other papers ($n = 28$). Anecdotal and distributional papers did not always refer to *X. laevis*, despite citations to the contrary. This was in part because of

taxonomic adjustments that have only been resolved recently (see above) and partly because citations often referred to overland movements in *Xenopus*, without specifying the species. Lastly, documents were classified according to whether they were reporting on invasive ($n = 16$) or indigenous ($n = 40$) populations. The network visualisation discriminates between citing and cited publications.

## RESULTS & DISCUSSION

### Evidence for overland dispersal

There is overwhelming support in the literature that *Xenopus laevis* moves overland from 18 papers in both native (eight papers) and invasive (10 papers) ranges. Most information comes from anecdotal observations (12 papers), where observers directly witnessed animals moving overland. Some reports (four papers) relate to distributional evidence that *X. laevis* had reached isolated ponds, and must have moved overland in order to do so. Although other explanations are possible (e.g., people or would-be predators deliberately or inadvertently moving adults, tadpoles or eggs), in the light of the support from direct observations, it seems more reasonable to accept that these distributional inferences do relate to overland movements. Similarly, mark-recapture studies (three papers) assume no other agent of overland dispersal is involved. In all three cases this appears more reasonable as time between captures is constrained to periods without water directly linking the sites, although see comments below on *Measey & Tinsley (1998)*. The telemetric study (*Eggert & Fouquet, 2006*) was able to follow animals out of ponds and observe them moving overland. Additional reports that refer to other species are covered below.

Most observational studies report the simultaneous movement of multiple individuals, often in very large numbers. Only three studies report observations of a single *X. laevis* (*Lobos & Garín, 2002*; *Eggert & Fouquet, 2006*; *Vanschoenwinkel et al., 2008*). Several anecdotes refer to spectacular mass overland movements of *X. laevis* in native and invasive populations (reported by *Channing, 2001*; *Crayon, 2005*; *Hewitt & Power, 1913*; *Lobos & Jaksic, 2005*). These examples all have the drying of an impoundment in common, where animals appear motivated to move by reduction in water level but notably do not wait until there is no water, instead leaving when levels are very low. There is a single account which suggests that such mass movements do not occur only as impoundments dry: *Wager (1986)* comments on large numbers of animals moving overland after heavy rains. The best documented account of mass overland movement comes from the observations of Gabriel Lobos who reported on movement in an invasive population of *X. laevis* in Chile. He noted that water levels had dropped to 5–15 cm (from several metres deep when the impoundment was full: *Lobos & Jaksic, 2005*). The animals that moved were in good condition, with no apparent sex bias (although no juveniles were seen moving) and estimated to number several thousand. A previous estimate of population for the same impoundment suggested that two years earlier numbers may have been as high as 20,000 (*Lobos & Measey, 2002*). Mass movements when water-bodies dry are also reported in South Africa, resulting in large amounts of associated road-kill (N Passmore, pers. comm., 2016). One noteworthy observation is that when moving *en masse*, the animals

form an unbroken chain (*Lobos & Jaksic, 2005*; *Weisenberger, 2011*). This might reflect the lead animals being stimulated by olfaction (*Savage, 1961*), while those following cannot see their leaders (see *Elepfandt, 1996*) and may not obtain olfactory cues, therefore trying to remain in physical contact with them. Perhaps unsurprisingly, anecdotes of smaller numbers of *Xenopus* moving have also been recorded when impoundments are drying (e.g., *Loveridge, 1976*).

Records of mass movements suggest that entire populations move (*Channing, 2001*; *Crayon, 2005*; *Hewitt & Power, 1913*; *Lobos & Jaksic, 2005*), but no reports have specifically tested this idea. In the cases where impoundments have dried and when burrowing into the substrate is not an option, it may be safe to assume that all individuals were forced to make overland movements. The coordinated onset of mass movement from drying puddles in *X. laevis* (*Lobos & Jaksic, 2005*) receives support from other synchronised behaviours such as air breathing (*Baird, 1983*). The only study that estimated the frequency of individuals moving overland suggests a surprisingly high proportion of the population. *Measey & Tinsley (1998)* report movements between capture sites (which necessitated overland movements) for 21% of individuals captured more than once in and around the Afon Alun, South Wales. At the other locality (Dunraven) this was as high as 36% although it is not clear whether animals had actually moved overland due to the existence of subterranean aquifers. Some authors mention movements between flowing and still water-bodies (*McCoid & Fritts, 1980*; *Measey & Tinsley, 1998*). *Measey (1997)* reports on recaptures from invasive Welsh populations and states that recaptures in ponds very close to a river were most common when the river was not flowing (see also *Measey & Tinsley, 1998*; *Tinsley et al., 2015*). This appears to suggest that these individuals were using permanent ponds mostly when the river dried. Interestingly, subsequent studies in the same area suggest that these movements decreased over time as the population waned (*Tinsley et al., 2015*). This may indicate that movements are driven, at least in part, by the existence of populations with high densities (see also *McCoid & Fritts, 1980*). *Measey (1997)* further notes that movements from river to ponds "...would have to be overland, and in the cases of FP and TFP [abbreviations of pond names], obstacles including vertical walls (up to 0.5 m) and dense hedgerows would have had to be traversed. Some of the animals caught were noted to have heavy scarring of dorsum and ventrum, consistent with movement over such terrain." This suggests that *X. laevis* are able to overcome modest obstacles in their path, in order to gain access to water-bodies. This concurs with observations in South Africa where walls and thick vegetation are regularly traversed (J Measey, pers. obs., 2014, also see *Schramm, 1987*). Similar observations have been made in other invasive populations where it was inferred that individuals had to move up steep walls and slippery slopes (R Rebelo, pers. comm., 2016). It would be of interest to determine whether these distributional inferences are accurate before considering building barriers to prevent dispersal in invasive populations of *X. laevis*. For example, a 0.7 m high concrete wall failed to keep *X. laevis* out of a *X. gilli* breeding pond (*Fogell, Tolley & Measey, 2013*; *Picker & De Villiers, 1989*).
**Table 1   Overland distances moved by *Xenopus* species recorded in the literature.**

| Reference | Species | Number of individuals | Distance reported (km) | Data type | Population |
|---|---|---|---|---|---|
| *Loveridge (1953)* | *X. borealis* | Unspecified | 0.45 | Anecdotal | Indigenous |
| | *X. muelleri* | >14 | 0.9 | Anecdotal | Indigenous |
| *Inger (1968)* | *X. muelleri* | 1 | 0.02 | Anecdotal | Indigenous |
| *McCoid & Fritts (1980)* | *X. laevis* | Unspecified | 0.8 | Anecdotal | Invasive |
| *Picker (1985)* | *X. gilli* | 11 | 0.9 | Mark-recapture | Indigenous |
| *Wager (1986)* | *X. laevis* | Thousands | 1.0 | Anecdotal | Indigenous |
| *Measey & Tinsley (1998)*[a] (*Measey, 1997*)[a] | *X. laevis* | 55 (21% of recaptures) | 0.2 (within 48 hrs) 0.75, 1.5 & 2.0 (direct distance) | Mark-recapture | Invasive |
| *Lobos & Garín (2002)* | *X. laevis* | 1 | 0.04 | Anecdotal | Invasive |
| *Lobos & Jaksic (2005)* | *X. laevis* | Hundreds | 0.1–0.18 | Anecdotal | Invasive |
| *Eggert & Fouquet (2006)* | *X. laevis* | 1 | 0.08 | Mark-recapture | Invasive |
| *Faraone (2008)* | *X. laevis* | Unspecified | 0.48 | Distributional | Invasive |

**Notes.**
[a] These sources report the same data.

## Distances moved overland by *X. laevis*

Reports of distances moved overland by *X. laevis* in the literature vary from 40 m to 2 km (Table 1) and are comparable to distances travelled by other terrestrial amphibians (*Smith & Green, 2005*). The distribution of distances reported appears to suggest that shorter distances are more unusual, very unlike a dispersal kernel for most anurans (see for example *Smith & Green, 2005*). However, a closer examination of the data sources reveals that most are anecdotal and are thus more likely to record longer movements, while empirical studies record both long and short distances. Distances of distributional displacements are in general accordance with those measured by empirical studies (Table 1), but both suffer from a lack of information about conditions during which movements occur. For example, *McCoid & Fritts (1980)* refer to sheet flooding facilitating the movement of juvenile *X. laevis* 800 m in San Diego County, USA. But *Lobos & Jaksic (2005)* specified that movements of nearly 200 m were made across dry ground near San Diego, Chile. Thus, it is hard to treat distances reported in the literature comparatively, as they may relate to quite different scenarios, with respective distances estimated in different ways. *Measey & Tinsley (1998)* and *Measey (1997)* who report the longest distances for overland movement of *X. laevis* of up to 2 km, provide straight line distances between capture sites, whereas distances actually travelled could have been much longer. However, this total distance could have included use of a river, making the maximum distance moved overland not 2 km but approximately 450 m, if dispersal events occurred through the flowing river. Most distances reported in the literature do not provide any indication of how they were estimated. Despite these issues, it is clear that *X. laevis* is able to move substantial distances overland. There is very clear evidence from one study that *X. laevis* are capable of moving at least 0.45 km overland (*Measey & Tinsley, 1998*), and more ambiguous evidence from two others indicating distances up to 1 km are possible (*McCoid & Fritts, 1980*; *Wager, 1986*).

### Is overland movement nocturnal, seasonal or weather-dependent in *Xenopus laevis*?

When weather is mentioned, nine out of 12 authors note that overland movements occur during or shortly after rain (*Channing, 2001*; *Du Plessis, 1966*; *Eggert & Fouquet, 2006*; *Fouquet & Measey, 2006*; *Hey, 1949*; *Loveridge, 1976*; *McCoid & Fritts, 1980*; *Picker, 1985*; *Wager, 1986*). But movement does not appear to be confined to wet periods or during rain-showers as three studies specified that conditions were dry for mass and single overland movements (*Hewitt & Power, 1913*; *Lobos & Garín, 2002*; *Lobos & Jaksic, 2005*). In addition, I have observed a single *X. laevis* moving overland in the middle of austral summer without any apparent motivation from rainfall or drying habitats (J Measey, pers. obs., 2014 19h00, 28 January 2016, at Jonkershoek). Of the six papers that have specified the time of observed overland movement, five instances took place at night (*Crayon, 2005*; *Eggert & Fouquet, 2006*; *Lobos & Garín, 2002*) or during the evening (*Hewitt & Power, 1913*; *Lobos & Jaksic, 2005*). Conversely, *Loveridge (1976)* recorded all overland movements of *X. laevis* early in the morning. That *X. laevis* would not move overland during the middle of the day (or at least would not start a movement during the day) does not sound unreasonable for a species which is prone to desiccation away from water. The available literature thus suggests that overland movements may peak during wet periods, but are by no means confined to rain or wet seasons.

### Does *X. laevis* migrate?

Observations recorded in the literature suggest that *X. laevis* move into ponds at the onset of rains, not only from areas that might have dried up, but also as normal/regular movement between ponds (*Hey, 1949*; *Picker, 1985*). Clearly, if animals are aestivating out of water, such movements do not need a great deal of explanation, but *Hey (1949)* and *Picker* (*1985*—although it is not clear whether he refers to *X. laevis*, *X. gilli*, or both) appear to describe the movement of animals from one pool to another. *Hey (1949)* specifically interprets these movements as a migration to breed in temporary waters, and this is repeated in correspondence reported by *Mahrdt & Knefler (1973)*. This record is of interest as Hey extends his observation to include "defined migration routes" for mass movements that occur at night in damp or cold weather. In addition, Hey notes that these routes result in mass mortalities when interrupted (the example given is the construction of a new barn; *Mahrdt & Knefler, 1973*), a similar observation having been made by *Loveridge (1953)*.

Migration from permanent to temporary water-bodies for spawning would allow the spatial separation of adults from larvae which might otherwise be cannibalised (e.g., *Measey, 1998a*; *Measey et al., 2015*; *Schoonbee, Prinsloo & Nxweni, 1992*) as temporary waters are likely to have reduced densities of occupants. Similarly, temporary waters are likely to be high in nutrients, sometimes experiencing algal blooms and having reduced predator pressure, making them ideal habitats for developing larvae. Such observations and distributional inferences are available from other species (*Rödel, 2000*; *Thurston, 1967*), but for *X. laevis*, *Du Plessis (1966)* noted that ponds that were fertilised attracted frogs to move into them before any algae had time to grow. In accordance with many observations, the stimulus to move into temporary waters comes with the initial rains that fill them,

and this is often combined with immediate egg laying (e.g., *Balinsky, 1969*). A movement into a temporary water-body suggests a reciprocal movement upon drying conditions (see above), except that in many anecdotes it is not clear whether individuals have moved from other (presumably permanent) water-bodies, or simply aestivated in the dry substrate (*Balinsky et al., 1967*).

*Hewitt & Power (1913)* recount an anecdote indicating that *X. laevis* were aestivating in the mud of a pond, and that when this mud was moved to a new location the frogs continued to aestivate, only re-emerging from this new location following rains. Such particular observations have also been made elsewhere (A Channing, pers. comm., 2016). This suggests that animals do burrow into the mud of some temporary waterbodies, but this does not seem to be consistent, as *Hewitt & Power (1913)* also note. It is worth noting that *Crayon (2005)* suggested that *Tinsley & McCoid (1996)* reported migration of "...0.2 km in late spring to a spawning site," but the idea that this was a migration to a spawning site was an embellishment by Crayon. Fuller accounts of the same movement (*Measey, 1997*; *Measey & Tinsley, 1998*), simply refer to a movement from a permanent pond to a temporary one within 48 h. Other data suggesting migration in the Welsh studies imply that animals moved from the seasonal river into permanent ponds (see above), although this could be interpreted as movement due to drying of habitat. Of all citations given by *Crayon (2005)* suggesting breeding migrations, only *Hey (1949)* and Hey's comments in *Mahrdt & Knefler (1973)* actually state this. Although there is no reason to dismiss Hey's statements, since he clearly was very familiar with the biology of this species after raising animals at Jonkershoek for export (see *Van Sittert & Measey, 2016*), he offered no evidence of migration, be it anecdotal or empirical. Thus the literature provides a clear hypothesis that *X. laevis* may migrate to spawn, as many other anurans are known to (e.g., *Lizana, Márquez & Martín-Sánchez, 1994*), but it seems likely that this behaviour would be context dependent. In the majority of its indigenous and invasive ranges, water-bodies inhabited by *X. laevis* are anthropogenically created impoundments. Testing a hypothesis on migration in *X. laevis* would require a set of relevant, natural water-bodies.

## Overland movement in other *Xenopus* species

This review of the literature presents sufficient anecdotal and distributional data of other *Xenopus* species suggesting that many of these also move overland (Table 1). There are data that indicate movement during dry periods in *X. borealis*, which suggests that mass movements also occur in other species (*Weisenberger, 2011*). Perhaps unsurprisingly, there are other anecdotal observations of mass movements for *X. borealis* (*Loveridge, 1953*), as well as *X. muelleri* (*Loveridge, 1953*; *Thurston, 1967*). Movements overland outside of rainy periods also exist for *X. tropicalis* (*Rödel, 2000* and references therein) and suggest that, like *X. laevis*, other *Xenopus* species move throughout the year irrespective of rains. To date, there is no reason to suspect that *X. laevis* moves any further overland than any of its congeners (see Table 1), despite its larger size. However, there are no suggestions that any congeners migrate to breed, which is perhaps not surprising, given that there is only a single suggestion of this happening for *X. laevis* (*Hey, 1949*). Thus, none of the movement for other *Xenopus* species appear to contradict the findings here for *X. laevis*, prompting

the question of whether any *Xenopus* species might be expected to be substantially different in their overland movement patterns? One species, *X. longipes*, stands out as, within the genus, it appears to be an example of a taxon which exhibits extreme adaptation to an aquatic lifestyle. Moreover, it is found in a single, hydrologically closed locality, namely Lake Oku, and no specimens have ever been found outside this lake, despite a recent increase in research interest in this species. As residents of a volcanic lake in the Cameroon highlands, it seems unlikely that this species would ever have experienced a drying habitat that might have prompted overland dispersal. Similarly, a lack of food and conditions prompting mass mortality events did not involve individuals leaving Lake Oku (*Blackburn et al., 2010*; *Loumont & Kobel, 1991*)

## Use of literature

Analysis of the use of literature allows an overview into the importance of this topic. The majority of studies (26 papers on *X. laevis* and two on other *Xenopus* species) which were found in the literature search did not have data on *Xenopus* overland movement (circles on right in Fig. 2). Those with original observations were mostly anecdotal in nature (13 papers on *X. laevis* and 6 on other *Xenopus* species are shown as squares on the left in Fig. 2), relaying information on instances where *Xenopus* have been observed moving overland. There was a clear trend over time for observations to move from anecdotes to distributional or mark-recapture data (four triangles and four stars, respectively), with interest in the topic clearly increasing as 60% of publications were published in the last 20 years (1995–2015) of the 102-year period. The majority of citations referred to publications with observations (curves above a direct connecting line between columns) or to reviews. However, there were several instances where curves below the line suggest that authors were citing publications without any data or observations. It is hoped that this review will help alleviate any past discrepancies in this respect. The network also showed that many of the citations refer to work that was conducted on invasive populations; to date, no empirical data exist on indigenous *X. laevis* moving overland, although both anecdotes and distributional inferences have been made. There is a clear need for empirical work in general, but in particular to fill the deficit identified here regarding indigenous populations of *X. laevis*. Limitations in the literature search were partially alleviated by adding expert knowledge of the literature, as well as using citations to publications from all articles identified. The existence of uncited literature, however, suggests that searches may not have been exhaustive and that other information on overland movement, particularly in other *Xenopus* species may shed further insight into this behaviour. There is clearly potential for new empirical studies on movements of *Xenopus* species within their native range.

## CONCLUSION

A review of the literature has shown that overland movements of *Xenopus laevis* have been found in both its native and invasive ranges. Although no empirical data exist for their native range, there is nothing to suggest that overland movements will be found to be less substantial or frequent than in their invasive range. Given the paucity of empirical

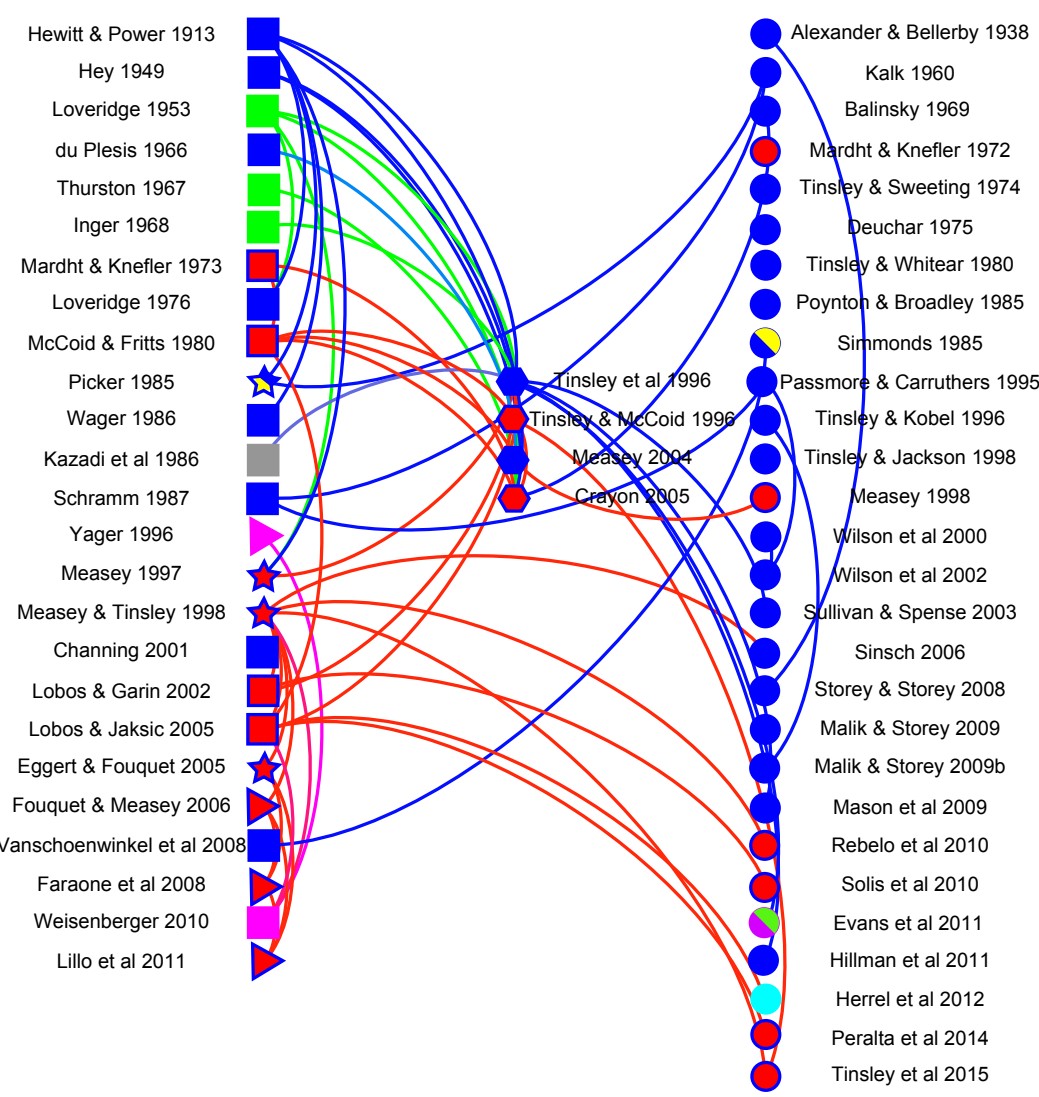

**Figure 2** Network visualisation for *Xenopus* overland movement literature. A network visualisation of literature mentioning overland movement in *Xenopus* using Gephi. Literature is sorted into that with data (left): anecdotal (squares), distributional (triangles), and mark-recapture (stars); literature reviews (middle: hexagons); and literature which does not have original data on overland movement in *Xenopus* (circles: right). Different species of *Xenopus* are denoted by different colours, and indigenous *X. laevis* (blue filled symbol) are differentiated from invasive populations (red filled blue symbol). Other species are coded as other colours: *X. muelleri* (green), *X. borealis* (pink), *X. gilli* (yellow), *X. fraseri* (grey), *X. clivii* (purple) and *X. tropicalis* (cyan). Curves connecting nodes denote the direction of the citation: above the line (right to left) or below the line (left to right). Nodes which are not connected represent literature which does not cite and has not been cited in relation to *Xenopus* movement overland. For complete references to the citations, please refer to Supplemental Information.

studies, distances moved appear to conform to those typical for other anurans, with large numbers of animals moving short distances and some individuals moving up to 2 km (direct distance). The literature does not appear to agree on whether overland movements are seasonal, although the majority of studies suggest that movements are more frequent when conditions are wet and they tend not to happen during the middle of the day. Lastly,

although suggested, there is currently no evidence in the literature to support the notion that overland movements are migrations to and from water-bodies for individuals to spawn. In addition to providing an overview on overland movements in *X. laevis*, this review also suggests that movement and breeding patterns for *X. laevis* may be similar to other members of the genus *Xenopus*. Although this review only mentions overland movement in six of 29 currently described species (*Frost, 2016*), lack of reports for the other species probably relates to a lack of research.

## ACKNOWLEDGEMENTS

I would like to thank those people who helped me obtain literature: Marié Theron, Alan Channing, Ed Stanley and Dennis Rödder. Thanks also to Brent Abrahams for help with the Gephi analysis. Lastly, I thank members of the MeaseyLab (especially André de Villiers and Ana Nunes), Thalassa Matthews, Donald Kramer and the INVAXEN group for fruitful discussions and comments on an earlier version of this manuscript.

### Funding

This research was funded by the National Research Foundation of South Africa (NRF Grant No. 87759), and the DST-NRF Centre of Excellence for Invasion Biology. The funders had no role in study design, data collection and analysis, decision to publish, or preparation of the manuscript.

### Grant Disclosures

The following grant information was disclosed by the author:
National Research Foundation of South Africa: 87759.
DST-NRF Centre of Excellence for Invasion Biology.

### Competing Interests

John Measey is an Academic Editor for PeerJ.

### Author Contributions

- John Measey conceived and designed the experiments, performed the experiments, analyzed the data, contributed reagents/materials/analysis tools, wrote the paper, prepared figures and/or tables, reviewed drafts of the paper.

### Data Availability

Literature used in the systematic review has been supplied as a Supplemental File.

### Supplemental Information

Supplemental information for this article can be found online at http://dx.doi.org/10.7717/peerj.2474#supplemental-information.

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
