# Peer review of "Overland movement in African clawed frogs (Xenopus laevis): a systematic review"

_PeerJ, doi:10.7717/peerj.2474_

## Round 0.1 · original submission · Major Revisions

Overview

I was requested to consider an Appeal of the Rejection decision by PeerJ for a manuscript submitted by John Measey that consists of a systematic review of overland movements in Xenopus.

Reviewer 1, who has substantial relevant publication experience, recommended rejection based on the inadequacy of the literature search, excessive description of the literature search process and citation pattern, lack of analysis of the biological questions, and an apparent failure to conform to the scope of the journal. Reviewer 3, whose research is somewhat more distant, also recommended rejection because it is a literature review and because the literature search was inadequate and the findings were poorly organized. Although Reviewer 3's report was brief, the feedback and justification for these comments were substantially enhanced by comments inserted on the pdf. In contrast, Reviewer 2, who is closest to the research area, recommended only minor revisions, largely clarification of wording, and he specifically mentioned the validity of the interpretation of the literature. This positive viewpoint is, however, confounded by a substantial COI, including a shared project and at least 6 co-authorships (Web of Science, assuming John Measey and G. John Measey are the same person), including one as recently as 2015. The Academic Editor followed the recommendations of Reviewers 1 and 3, rejecting the manuscript on the basis of an inadequate literature review, excessive emphasis on how the papers were selected and insufficient attention to the biological questions.

Is a systematic literature review appropriate for PeerJ? The answer seems to be 'yes' based on the specific guidelines for systematic reviews and meta-analyses listed as item 4 under 'Discipline specific standards' under 'Policies and Procedures'. However, authors, reviewers and editors might be forgiven for not understanding this because literature reviews are explicitly excluded by the Aims and Scope statement, without any indicator of exceptions. The term 'systematic review' comes from evidence-based medicine and is not widely used in ecology and behavior. Indeed, I was unfamiliar with the term and thought it referred to taxonomic studies. A quick search of Web of Science shows that I could have become acquainted with the term before now, with a few conservation and other applied ecology papers starting about 2002 (earlier papers with this term being mostly taxonomic as I had supposed). Nevertheless, it might be appropriate for author to briefly explain his approach with appropriate citations rather than to assume that every reader will recognize the term.

Is the literature review adequate? The reports of Reviewers 1 and 3 may overstate the inadequacy of the literature review. The author indicated (L98) that he also carried out searches on Web of Science and Scopus which yielded fewer articles and none that had not been found using Google Scholar and that he followed up citations in the literature he did find. As an active researcher in this field, it seems unlikely that he would be unaware of any except the most recent work in this area. Nevertheless, I agree that 'overland' as a search term does seem potentially limited and the suggestion of Reviewer 3's inserted comments to consider common names of the species and other terms such as terrestrial does seem valid. Thus, the quality of the review could be improved with a relatively small effort but seems unlikely to make a large difference in the results.

Is it appropriate to include the formal presentation of the literature search, including Figure 1? The answer is 'yes'. Fig. 1 is a requirement for publishing a Systematic Review in PeerJ according to the Discipline Specific Standards and in many medical journals according to the Wikipedia page I consulted to understand the concept. However, the author could explain why this is integral to the approach and indeed a citation seems appropriate on L113, given that he used a standard formulation from a specific organization web page.

Is it appropriate to include the Network visualization, including Figure 2? I am not sure of the answer to this question. I did not find any explicit statement that this should be part of a systematic review. Unlike the reviewers, I found it interesting, but felt that the author needed to explain how this visualization contributed to achieving his objectives. Furthermore, Figure 2 itself is not fully explained and the maximal contribution of this analysis was not achieved. (I inserted comments on the pdf concerning my views on these issues.)

Is the analysis of the results of the review adequate? Here, I think that the answer is a clear 'no', but the deficiency is not sufficient to reject the manuscript but rather indicative of the need for major revision. In particular, the review lacks a critical evaluation of the evidence presented and it lacks a quantitative synthesis. For example, there is no explicit statement of the number of anecdotal, inferred and empirical observations of overland movements have been reported. (As an aside, I find the distinction between inferred and empirical disconcerting because inferential evidence is empirical too.) There is no separation of the studies into those based on a single observation from those citing repeated from multiple occasions. There is no statement of what an example of overland movement consists of and what alternative explanations need to be considered for a valid explanation. I have indicated a number of places in the manuscript which need such a critical and quantitative approach and some additional ones are raised in the next section of my comments, but the author should consider this issue throughout.

Is the presentation adequate? Again, I agree with Reviewers 1 and 3 that the presentation is weak. The information is poorly organized, despite the value of the specific questions as a natural organizational framework.

For example, the first topic in the Results and Discussion is 'Literature use', an unclear concept which might be considered to relate to Fig. 2. However, the section seems to be address Question 1, at least in part. A reader looking here for an answer to the question 'What is the evidence for overland dispersal', would not be find a critical, quantitative summary in this section.
As another example, the section intended to answer the question 'What distances are moved overland?' does not answer the question. It only provides a distance from one study and immediately casts doubt on its validity. It refers to a table that is ordered chronologically and does not provide a quantitative synthesis of the material listed there. The reader is provided no information regarding the type and strength of evidence in those studies. I would have expected a critical evaluation of the evidence both that animals actually moved and the distances involved and then a summary of maximal distances in relation to the quality of evidence. Comments related to the difficulty of drawing conclusions from some reports can be included, but the present report seems to use the challenges as an excuse for failing to provide a clear answer. I would look for a strong development of the evidence leading logically to a summary at the end of such as paragraph such as, 'Thus, we have very clear evidence from three studies that Xenopus are capable of moving at least 0.5 km overland and more ambiguous evidence from five others indicating distances up to 2 km are possible.'
Subsequent sections have titles related to the a priori questions, but the order is inexplicably reversed from the order in which they are raised in the Introduction. The flow of ideas from data to conclusions is generally not clear. In addition, an unacceptable number of grammatical and spelling errors interfere with reader comprehension. Many, but far from all, of these points are indicated in my comments directly on the attached pdf.
Despite these difficulties, weaknesses of presentation normally call for major revision rather than rejection.

In summary, my opinion is that this manuscript has substantial weaknesses but that these are not the result of fatal flaws in the study design or approach but of the depth of critical analysis and quantitative synthesis and of clarity of presentation. The author should be therefore offered the opportunity for a major revision, with the recognition that this would involve rethinking and reorganizing most of the manuscript.

· Appeal

Appeal

I would like to appeal against this decision - which itself appears to be based on a flawed understanding of a systematic review.

The editor and reviewers don't feel that Figure 1 is necessary - that they are ignorant that it is a requirement of a systematic review in PeerJ is unfortunate, but says less about a failure in my submission that their own reviewing/editing skills. The editor complains that "Much effort is focused on the logical [sic] of how papers were chosen for inclusion and, instead, I would have liked to have seen more emphasis on the biological question". Again, this displays a fundamental ignorance of a systematic review, which the editor should be made aware of before taking on the handling of a systematic review. The reviewer complaints (with which the editor agreed) that the review is flawed as there is only one database used is a sad reflection on their ability to read a manuscript (3 databases were used).

I know that it is not PeerJ policy to make sure that their editors are aware of the journal's requirements for systematic reviews, but perhaps it should be? I'd suggest that systematic reviews come with an extra sheet of information on the reviewers' copy, otherwise other authors will have to put up with inappropriate comments from reviewers and editors alike.

With best wishes,

John


· · Academic Editor

Reject

While the subject matter of this manuscript is interesting, the methodology is flawed. I agree with Reviewers 1 and 3's comments with respect to problems associated with the way the literature search was conducted. Unfortunately, addressing this would require the author to start from the beginning. Further, I agree that the Figures do not support the science but, rather, are more philosophical in nature. Much effort is focused on the logical of how papers were chosen for inclusion and, instead, I would have liked to have seen more emphasis on the biological question. If the author was inclined to conduct this review again more thoroughly, I hope that the comments of all three reviewers will be of use.

Reviewer 1 ·

Basic reporting

The article is written in plain English.
I did considered table 1 to be somewhat confusing however as it is not clear what the author means by 'literature which reports the same data'. I also noted a spelling error in the associated legend.
The figures betray the fact that article does not seem to conform the scope of the journal and were not at all what I was expecting based on initially reading the manuscripts abstract.
There was no attempt to compare the data reviewed in any rigorous manner, nor produce the figures I would have expected to see associated with this article.

Experimental design

Unfortunately I believe the manuscript fails to meet the stated scope of the journal in this regard. There is no attempt at analysis of the biologically meaningful data collected in the (very rudimental) literature search. The questions raised were not rigorously investigated, nor answered.

Validity of the findings

The author uses a single two-word ‘search term’ to query one single database and ‘harvests’ a list of references, then proceeds to pare down the reference list in excruciating detail based on seemingly logical criteria with which to assess the relevance of the literature. Rather than perform any analysis on the data to answer any of the questions raised in the introductory sections of the manuscript, the author speculates extensively on the meaning of all of the data and presents graphical representations of how the actual literature has travelled through literary space.

Additional comments

This seems to be a reasonable, though still quite basic, review of the literature in order to answer some interesting questions. Rather than painstakingly describe the review, I would expect to see analysis of the data pertaining to the biologically relevant questions and reporting these results before considering a manuscript for any biology journal.

·

Basic reporting

Basic reporting is good. There are just a few areas where things could be clarified as listed below.

Experimental design

This is a review so there is no experimental design.

Validity of the findings

The findings are valid and the interprettion of the literature search seem solid.

Additional comments

line 17: rephrase, sounds awkward

line 40: continue sounds odd in this context. Maybe use 'complete'

line 42: larvae, prompting...

lines 47-48: difficult to understand ... rephrase.

line 65: an alternative explanation is that this could simply be phylogenetic inertia since their ancestors were terrestrial.

line 103: delete 'the given'

line 117: was classified?

line 189: here it is not clear? rephrase

line 205: to move up steep ...

line 206, sentence starting with 'to build barriers' makes no sense. rephrase.

line 217: add year after Hey

line 244: imply

lines 249-256: a bit long ... you lost me here as a reader.

line 294: your sentence is not finished ...

·

Basic reporting

Figure 1 & 2 not relevant to the article.
Did not contain all results- it was not clear how many empirical versus anecdotal studies relevant to overland movement in Xenopus were found

Experimental design

This was not original research- it is a literature review.
The investigation (literature review) was not rigorous enough. one search engine was used with one set of key words.

Validity of the findings

Literature review (see comments above)

The investigation was not rigorous enough and the results/discussion were not written in a very organized/ coherent way.

Additional comments

see annotated PDF

---

## Round 0.2 · Minor Revisions

This manuscript has been substantially improved. Only a few substantive issues and a small number of grammatical errors remain.

L10, 76. The Abstract and Introduction differ in the framing of question 1. I feel that the question as stated in the Introduction (evidence for overland movement) is broader and more primary than the narrow question in the Abstract (differences in movement between native and invasive ranges) and should be the one addressed.

L12, 79. I think the fourth question would be clearer if migration was specified as breeding migration.

L121-124. I still have a problem with two of your terms for categorizing the types of studies. 'Anecdotal' and 'review' present no problem of name or definition.
The term 'inferred' initially seemed acceptable, though 'inferential' would be grammatically more parallel to 'anecdotal'. (These are adjectives with 'evidence' as the implied noun.) However, there is a problem in that the third category, if I understand correctly, also uses inferred evidence (see below). Furthermore, the definition of inferred is not acceptable because it uses the term in its own definition. I suggest that the definition is something like 'publications with indirect evidence such as the appearance of frogs in isolated ponds from which they had previously been absent'. If all the papers in the inferred category have this type of evidence, you can remove the 'such as'.
The term 'empirical' really does not work for the third category because it does not distinguish this category from anecdotal and inferred/inferential, which are also empirical. Furthermore, the definition fails to clarify the point because it refers to the motivation of the researcher rather than the nature of the evidence. I had trouble imagining what kind of experiments would be involved and speculated that maybe researchers may have released frogs at different distances from water or counted frogs moving on land under different environmental conditions. I tried to look up the 4 articles to see if I could understand the studies involved and suggest a better term. Unfortunately, I was not very successful. Two of the articles were not available to me online, and I did not attempt to find the thesis. However, I did scan Picker 1985. This was not a true experiment (i.e. manipulation of a hypothesized causal variable with a control) as implied. It was an observational study that used marking to identify frogs that had moved between ponds. Thus, the logical structure of the evidence is similar to the inferred category: in both cases, movement is inferred without direct observation by the appearance of frogs in places where they had not been not previously.
Could I suggest the following?
Observational: anecdotal reports of frogs moving in terrestrial habitats
Distributional: occurrence of frogs in water bodies separated by terrestrial habitats from potential source populations
Mark-recapture: occurrence of marked individuals in water bodies separated by terrestrial habitats from water bodies in which the frogs had been marked
Reviews
If these proposed categories and definitions fail to properly capture the range of articles included or if you do not like them for some other reason, perhaps they and my comments may inspire you to find alternative terms that are exclusive from each other and linked with definitions that are precise and clear for anyone repeating or paralleling your study.

L134. You need a subhead related to your questions: 'Evidence for overland dispersal'
L138. You need to provide a summary sentence about the type evidence in your third category (empirical/mark-recapture). For each category, there should be a critical discussion of the nature of the evidence and potential for misinterpretation. For example, how certain is it that the intervening terrestrial habitat never floods enough to provide an aquatic pathway? The sentence starting 'Additional reports . . . ' breaks the logical flow and should be the final sentence of the paragraph.
L143-194. These two paragraphs require better organization. The number of individuals moving together is one topic. The context (drying ponds, river flow reduction, population density) is another topic, probably more appropriate under the subhead concerning seasonality. Barriers to movement (or lack thereof) is a third topic.
L143. Improve the topic sentence of this paragraph to focus on the main message, for example: Most observational studies report the simultaneous movement of multiple individuals, often in very large numbers. (Then continue with the summary of reports for multiple individuals, leaving the single individuals for the end, but possibly mentioning anything distinctive about those two cases.)
L155. 'in their thousands' is colloquial. As an alternative, '. . . and estimated to number several thousand individuals' (more precision would be desirable if provided by the original source).
L155. impoundment, not dam
L169. Having discovered synchronous air breathing in fish (Kramer & Graham 1976, Copeia) before Baird reported it in Xenopus and having followed the literature to some extent since, I really doubt that it has any bearing on aggregation during overland movements.
L180. decreased, not reduced
L193. Ungrammatical sentence. Remove or modify 'constructed'.
L219. weather-dependent (hyphen and spelling)
L232. Since you are drawing a conclusion based on previous information, ' . . . literature thus suggests . . . '
L279-280. Reconsider the suggestion that there would be no advantage to migration in areas where all water bodies are ephemeral. Water bodies may vary in their duration, depth, food availability, cover, predator abundance. Even if they were all the same, density-dependent effects in mating and adult or larval foraging could provide an advantage to dispersal.
L304. This is not clear. Do you mean that there was a mass mortality in the lake and the frogs nevertheless did not leave? Does this refer to the same lake in the Cameroon highlands?
L314. 102-year period (hyphen)
L334. 2 km (space)
Fig. 1. Consider adding a reference to the caption to credit the source of the figure format.
Fig. 2. Is Eggert and Fouquet 2005 as in references and text the same as Eggert and Fouquet 2006 in the figure?
Table 1. I think the footnote would be clearer if you wrote '*These sources report the same data.'

---

## Round 0.3 · accepted · Accept

The manuscript is now suitable for publication. I think that the new terminology for classifying publications make the text clearer, e.g., the need for mark-recapture studies rather than empirical studies in the native range. Unfortunately, there are a few places where the old terminology has persisted. I believe that these can be fixed during the preparation for publication for publication rather than requiring another round of revision. I also noted two other minor errors.

Replace 'inferred' by 'distributional':
Table 1, Figure 2 caption

Replace 'empirical' by 'mark-recapture':
L210, 211, 287, 328, 329, 335, 338, 340, Table 1 (3 places), Figure 2 caption

L223. Add a period after 'overland'
L338 no data exist (plural)